# The Influence of an Extended π Electron System on the Electrochemical Properties and Oxidizing Activity of a Series of Iron(III) Porphyrazines with Bulky Pyrrolyl Substituents

**DOI:** 10.3390/molecules28207214

**Published:** 2023-10-22

**Authors:** Tomasz Koczorowski, Tomasz Rębiś

**Affiliations:** 1Chair and Department of Chemical Technology of Drugs, Poznan University of Medical Sciences, Rokietnicka 3, 60-806 Poznan, Poland; 2Institute of Chemistry and Technical Electrochemistry, Poznan University of Technology, Berdychowo 4, 60-965 Poznan, Poland; tomasz.rebis@put.poznan.pl

**Keywords:** iron, porphyrazine, electrochemistry, spectroelectrochemistry, catalysis

## Abstract

The present study investigates four iron(II/III) porphyrazines with extending pyrrolyl peripheral substituents to understand the impact of introduced phenyl rings on the macrocycle’s electrochemical and spectroelectrochemical properties as well as their activity in oxidation reactions. The electrochemical studies showed six well-defined redox processes and quasi-reversible one-electron transfers—two originating from the iron cation and four related to the ring. Adding phenyl rings to the periphery increased the electrochemical gap by 0.1 V. The UV–Vis spectra changes were observed at the applied potential of −1.3 V with the presence of additional red-shifted bands. The oxidizing studies showed increased efficiency in the oxidation reaction of the reference substrate in the cases of **Pz1** and **Pz2** in both studied oxygen atom donors. The calculated reaction rates in *t*-BuOOH were 12.0 and 15.0 mmol/min, respectively, for **Pz1** and **Pz2**, compared to 6.4 for **Pz3** and 1.8 mmol/min for **Pz4**. The study identified potential applications for these porphyrazines in mimicking cytochrome P450 prosthetic groups for oxidation and hydroxylation reactions in the future.

## 1. Introduction

Among the extensive group of porphyrinoids, porphyrazines (Pzs) constitute a distinct class that differs in chemical structure from natural porphyrins and artificial phthalocyanines. Unlike the latter, Pzs feature azamethine bridges that fuse pyrrole rings together, as opposed to isoindolinyl in phthalocyanines. Porphyrazines exhibit unique physicochemical and electrochemical properties, which can be tailored through the exchange of the metal cation within the core or through peripheral modifications at the β positions of the pyrrolyl substituents, involving aliphatic, aromatic, sulfur, oxygen, and nitrogen moieties [1,2,3,4,5]. By introducing transition metal cations into the core, Pzs can be employed for various electrochemical and catalytic applications, such as the oxidation of linear and cyclic alkenes, and the photocatalytic degradation of organic dyes found in industrial waste streams, including Rhodamine B and X3B dye (Reactive Brilliant Red) [6,7,8,9,10,11]. Over the past two decades, extensive research has been conducted on the electrochemical and catalytic properties of porphyrazines, particularly in terms of their sensing capabilities [12]. Iron(II/III) complexes of porphyrazines are the most commonly studied [5,13]. In 1992, Fitzgerald et al. conducted various studies, revealing significant differences in the physicochemical properties of Ps, Pcs, and Pzs, all of which possessed the same peripheral substituents and iron(III) cation within a macrocyclic core [14]. It was suggested that porphyrazines exhibit stronger σ-donor and π-acceptor characteristics compared to porphyrins. Electrochemical studies indicate that, similar to phthalocyanines, porphyrazines exhibit a positively shifted redox potential, differing by 0.4 V from their porphyrin counterparts. The conjugative pathway of porphyrazines consists of 18 π electrons, enabling π-π interactions with other molecules and carbon-based nanomaterials, like graphene or carbon nanotubes, facilitating the stable adsorption of Pzs onto material surfaces. It has been established that expanding the π electron system in porphyrinoids increases the binding energy with carbon nanotubes due to additional π-π interactions [15].

The chemical structure of peripheral substituents in porphyrazines can profoundly influence their electrochemical properties, catalytic capabilities, and aggregation tendencies [16], and may even result in the formation of atropoisomers due to bulky substituents restricting rotation around single C–C, C–O, or C–N bonds. While β,β-substituted, fused, heterocyclic ring Pzs have been widely studied [17,18], limited data exist regarding those modified by peripheral heteroaromatic group attachments. An example is a two-core Pz reported by Luo et al. [19], substituted with trimethyl-3-thienyl groups on its periphery. Other examples originate in our group. Over the past 15 years, our group has synthesized numerous Pzs bearing peripheral 2,5-dimethylpyrrol-1-yl, 2,5-dithienylpyrrol-1-yl, 2,5-di(biphenyl-4-yl)pyrrol-1-yl, 2-(1-adamantyl)-5-phenylpyrrol-1-yl, and 3,4-dihalide-2,5-dimethylpyrrol-1-yl moieties [20,21]. These Pzs have been thoroughly assessed in terms of their spectral, electrochemical, and catalytic properties. One study revealed that bulky 2,5-diphenylpyrrol-1-yl substituents on the periphery of iron(II) porphyrazine influenced its physicochemical properties, a finding confirmed by Mössbauer spectroscopy [16]. Other studies demonstrated that the iron cation’s valence and spin state could be easily tuned and were dependent on the work-up methods [22].

In the present study, a series of iron(III) porphyrazines, denoted as **Pz1**–**Pz4**, were synthesized with peripheral pyrrolyl groups substituted at the second, third, and fifth positions with methyl and/or phenyl groups. This synthesis employed a cyclotetramerization approach, utilizing both symmetrical and unsymmetrical maleonitriles. The objective of this study was to evaluate how expanding the electron π-system by adding phenyl groups, which have a positive inductive effect, influenced the electrochemical and catalytic properties of Pzs. These properties were assessed in the context of oxidation reactions involving the reference compound, 1,3-diphenylisobenzofuran (DPBF). The iron(III) porphyrazines with an extended π electron system could find potential applications in surface modifications of carbon-based nanomaterials for sensing and catalytic purposes.

## 2. Results and Discussion

Four symmetrical iron(III) porphyrazines possessing alternate peripheral systems consisting of pyrrolyl groups substituted in their second, third, and fifth positions with methyl and/or phenyl groups and dimethylamino substituents (Figure 1) were synthesized following the procedure described earlier [22]. In brief, proper unsymmetrical maleonitrile derivatives were subjected to a cyclotetramerization reaction with dimethylaminoethanol (DMAE) [23], and the obtained free-base porphyrazines were treated with FeBr_2_ in the presence of 2,6-lutidine dissolved in a toluene–THF mixture, according to the literature procedures [24]. Then, the synthesized crude products were purified using a dichloromethane/HCl two-phase system and column chromatography to obtain iron(III) Pzs–chloride complexes **Pz1**–**Pz4**. All macrocycles were characterized using common analytical techniques (MS MALDI TOF, NMR, and UV–Vis) and their purity was assessed by HPLC analysis. Their physicochemical characterization, as well as spectral analyses, have been described in detail in previously published papers [22].

According to the chemical structure of the obtained porphyrazines, a series of related macrocyclic compounds were synthesized, and differentiated in their expanding bulky periphery by introducing one, two, or three phenyl groups into pyrrolyl substituents (Figure 1). The proposed chemical modification led to the extension of the π-electron system, which influenced the electrochemical, spectroelectrochemical and catalytic properties of the obtained compounds.

### 2.1. Electrochemical Studies

Due to the presence of the transition metal cation with expected oxidation–reduction properties, the synthesized iron(III) porphyrazines **Pz1**–**Pz4** were electrochemically characterized by cyclic and differential pulse voltammetry. The electrode potentials of macrocyclic compounds were determined and assigned to specific redox processes. All electrochemical measurements were performed in a three-electrode system. A glassy carbon (GC) electrode was used as the working electrode, a platinum wire as the auxiliary electrode, and a silver wire covered with AgCl as the quasi-reference electrode. In the measurements of porphyrazines carried out in dichloromethane/0.1 M TBAP (tetrabutylammonium chlorate), ferrocene was additionally used as an internal standard. The mixture containing the solution of the appropriate porphyrazine and the electrolyte was saturated with nitrogen every time to remove dissolved molecular oxygen.

The cyclic voltammograms recorded at 50 mV × s^−1^ for porphyrazines **Pz1**–**Pz4**, having peripheral alternating pyrrolyl and dimethylamine groups, showed the presence of six redox peaks in each case (Figure 2). Based on the measurement of the signal width obtained by the DPV method, they were defined as one-electron processes [25]. The presence of six signals originating from oxidation–reduction processes was also previously observed by Arıcı et al. for iron(III) phthalocyanines [26]. Only four redox signals were detected in most cases of other synthetic porphyrinoids, like phthalocyanine and porphyrazine transition metal complexes. Two originated from the central metal cation (e.g., iron), whereas the other two were ascribed to the electronic processes within the macrocyclic ring [27]. In the present work, the observed elevated electrochemical activity is presumed to be related to the nitrogen-containing peripheral groups. The additional redox processes resulted from lone pairs of electrons or aromatic systems, contributing to the development of the general macrocyclic π-electron system.

The values of peak currents measured for porphyrazines **Pz1**–**Pz4** indicated a quasi-reversible nature of the processes, characterized by the ratio of the anodic *i*_a_ and cathodic currents *i*_c_. The obtained potential values were assigned to the appropriate redox processes (Table 1). The first two signals formed on the voltammograms were related to the reduction process in the macrocyclic ring, conventionally described as (**I**) Fe(I)/Pz(0)/Fe(I)/Pz(−1) and (**II**) Fe(I)/Pz(−1)/Fe(I)/Pz(−2), accompanied by the loss of two electrons. The next two came from oxidation–reduction processes involving the centrally coordinated iron cation, Fe^2+/3+^, which underwent redox reactions as a result of the change in electrode potential, described as (**III**) Fe(I)/Pz(0)/Fe(II)/Pz(0) and (**IV**) Fe(II)/Pz(0)/Fe(III)/Pz(0). In the case of the **IV** process, due to poor reversibility, the value of ΔE_p_ was not determined. Further potential change led to the occurrence of two subsequent oxidation processes related to the further acquisition of electrons by the porphyrazine macrocyclic ring: (**V**) Fe(III)/Pz(0)/Fe(III)/Pz(+1) and (**VI**) Fe(III)/Pz(+1)/Fe(III)/Pz(+2). Interestingly, in the case of **Pz4**, the determined ΔEp of process **VI** was below 50 mV, which may indicate the adsorption of the macrocycle on the surface of working electrode. Assignment of the oxidation/reduction signals **III** and **IV** to the centrally coordinated iron cation was associated with the cations of the d-block metals having a specific energy state. Nominally, this state, determined by the electrochemical method, is between the observed processes concerning the energy of the HOMO–LUMO orbitals of the macrocyclic ring of porphyrinoids. In the case of measurements of compounds **Pz1**–**Pz4**, this applies to processes **II** and **V**.

It is worth emphasizing that the presence of four signals of redox processes in the macrocyclic ring of the porphyrazines obtained indicates a strong electroactivity of the compounds as well as a highly reversible nature of the oxidation/reduction reaction. Such properties are particularly advantageous when designing potential amperometric sensors and electrocatalysts. The calculated E_1/2_ gap based on the potential difference between processes **II** and **V** increased by 0.1 V in the case of **Pz3** and **Pz4** compared to **Pz1** and **Pz2**. This may be a result of the absence of weak electro-donating methyl groups in the pyrrolyl substituents of **Pz1**–**Pz2** replaced by aromatic moieties in **Pz3**–**Pz4**.

### 2.2. Spectroelectrochemical Measurements

The spectroelectrochemical measurements enabled the assessment of changes in the oxidation states of porphyrazine systems equipped in peripheral alternating pyrrolyl and dimethylamino groups. The most interesting results were obtained for iron(III) complexes **Pz2** and **Pz4**, for which the formation of reduced forms of Pzs was observed in the negative potential range by the occurrence of new red-shifted bands next to Q bands (Figure 3 and Figure 4).

The electrolyzes of both **Pz2** and **Pz4** were performed at the applied potentials of −1.3 V and 1.0 V, where only changes in the UV–Vis spectra were observed. The negative potential value was closely related to the **III** redox process associated with a reduction in Fe^3+^ cation inside the macrocyclic core of porphyrazines, whereas the positive one was higher than the second oxidation processes (**VI**) of Pzs.

Considering the presence of an electrochemically active Fe^3+^ ion, the rationale for the appearance of new red-shifted bands during electrolysis at the negative potential can be explained by the formation of the Fe^2+^ complex of porphyrazines. Such changes in the UV–Vis spectra of iron(II/III) porphyrazines were observed earlier in the case of octaphenyl-substituted iron(III) porphyrazines where, in solvents with higher dielectric constant and/or donor properties (acetone, dimethyl-sulfoxide, pyridine, etc.), Fe^3+^ was reduced to Fe^2+^ with the formation of diamagnetic six-coordinated iron(II) complexes (L)_2_FePz [28], which could be noticed by changes in the UV–Vis spectra.

### 2.3. Oxidation Ability Studies

The last stage of experimental work consisted of experiments aimed at evaluating the potential oxidizing activity of synthesized macrocyclic compounds as systems capable of imitating the function of cytochrome P450 prosthetic groups in terms of oxidation and hydroxylation reactions due to the structural similarity between the obtained porphyrazines and heme.

Studies were carried out to assess the reactivity of **Pz1**–**Pz4** porphyrazines, using the example of the oxidation of 1,3-diphenylisobenzofuran (DPBF) as a reference compound. For this purpose, the literature procedure previously used in the singlet oxygen generation studies was adapted [29]. The DPBF oxidation process (Figure 5) is illustrated by a decrease in band absorption with a maximum at ca. 413 nm (Figure 6). For this reason, tests were carried out to measure the reduction in the absorption intensity of the mixture of DPBF and the appropriate iron(III) porphyrazine after the addition of an oxygen atom donor (OAD). Following the literature procedure for singlet oxygen generation studies involving DPBF as a singlet oxygen quencher, utilized in similar experiments by many research groups, reaction monitoring with a UV–Vis spectrophotometer was sufficient, and there was no need to use further analytical techniques, e.g., NMR or mass spectrometry. Moreover, additional analytical reaction monitoring would require the isolation of 1,2-dibenzoylbenzene as the product of DPBF oxidation. The obtained isolated amounts might be insufficient to record a spectrum with good quality.

Two most frequently indicated sources of oxygen were used in the following measurements: hydrogen peroxide and *tert*-butyl hydrogen peroxide. In the reactions, a two-fold excess of oxygen sources was used in relation to the oxidized substrate (DPBF). In order to reduce photodynamic processes and eliminate the influence of singlet oxygen on the measurement result, the reactions were carried out without access to light. *N,N*-dimethylformamide was used as the solvent. Before the measurements, the solvent was dried, and in order to deoxidize, it was saturated with nitrogen gas. During the oxidation process lasting 8 min, the absorbance in the range of 200–800 nm was measured every 2 min. After this time, the reaction was stopped in order to avoid additional effects, such as Fenton’s reaction, which could have arisen due to the degradation of the catalyst and the release of Fe^2+/3+^ iron ions. Each reaction was repeated three times and the results were averaged. In addition, two types of controls were performed, containing (i) DPBF and a solution of oxygen source (OAD) and (ii) a mixture of DPBF and the appropriate porphyrazine (without OAD).

It should also be noted that, in addition to a decrease in the absorption intensity of the DPBF band from λ_max_ = 413 nm, a decrease in the absorption of the Q bands of the analyzed porphyrazines was observed. The reason for these observed changes was the gradual decomposition of porphyrazines, caused by the destructive influence of hydroxyl radicals, formed as a result of homolytic cleavage of O-O bonds in OAD molecules. These occurring changes were previously observed by Su et al. [10] and Theodoridis et al. [6] in analogous studies of the oxidative capacity of iron(III) porphyrazine complexes and interpreted to be the destructive effect of the formed hydroxyl radicals on the porphyrinoid catalyst.

The obtained results showed that the most active catalysts in both utilized OADs were **Pz1** and **Pz2**, where the number of the phenyl rings in pyrrolyl groups was none or one, respectively (Figure 7 and Figure 8, Table 2). On the other hand, the increase in aromatic substitution in **Pz3** and **Pz4** caused a decrease in the degradation of the macrocyclic catalyst. Furthermore, comparing the two oxygen atom donors used, *tert*-butyl hydrogen peroxide was more efficient than H_2_O_2_ due to the formation of a high-active *t*-BuOO^●^ radical.

Regarding the usage of hydrogen peroxide as an oxygen atom donor, its use is associated with the production of high-active radical species, causing the substrate’s degradation in the catalytic reaction. Such activity can be used in water remediation, where the process is called catalytic wet hydrogen peroxide oxidation (CWPO) [8]. According to Su et al. [10] and Theodoridis et al. [6], when this oxygen atom donor is used in catalysis with iron(II) porphyrazine, the hydroperoxo-iron(III) complex of the macrocyclic compound is formed. Based on the reaction conditions, there are two competing redox pathways: heterolysis, which involves the transfer of two electrons, and homolysis, which involves the transfer of one electron, resulting in the cleavage of the O–O bond. When the conditions are acidic, a proton engages in the heterolysis of the O–O bond, giving rise to the formation of a transient high-valence iron-centered oxidizing species, denoted as Pz^●+^FeIV = O. It is worth noting that the electronic structure of N4-ligand complexes permits the stabilization of these transient high-valent intermediates. This is why high-valence iron species are frequently identified as ROS in biomimetic catalysis. Conversely, under neutral and alkaline pH conditions, homolysis of the O–O bond leads to the generation of hydroxyl radical species as ROS. Furthermore, the hydroperoxo complex of porphyrazine is transformed into a Pz radical, denoted as FeIII = O^●^ in this context [10].

## 3. Materials and Methods

### 3.1. General Synthetic Procedures of ***Pz1***–***Pz4***

The synthetic route of compounds **Pz1**–**Pz4** was performed following the subsequent procedure: Firstly, a free-base (2H) porphyrazine ligand (with the following particular amounts: Pz1: 84 mg, 0.09 mmol; Pz2: 48 mg, 0.043 mmol; Pz3: 85 mg, 0.063 mmol; and Pz4: 25 mg, 0.015 mmol) was added to a 25 mL round-bottom flask, along with FeBr_2_ (194 mg, 0.9 mmol for Pz1; 93 mg, 0.43 mmol for Pz2; 135 mg, 0.63 mmol for Pz3; and 32 mg, 0.15 mmol for Pz4), 2,6-lutidine (2 mL), and a toluene–THF mixture (1:1, 10 mL). The reaction mixture was stirred in reflux for 20 h. Next, the obtained green mixture was cooled to room temperature and evaporated to dryness. Subsequently, the solid was dissolved in dichloromethane (10 mL) and mixed with 1 M HCl (10 mL) for 30 min in a two-phase system. Then, the organic layer was separated, collected, and mixed with brine (10 mL) for 30 min (also in a two-phase system). After another separation, the organic mixture was dried with anhydrous MgSO_4_ and evaporated to dryness using a rotary evaporator. To remove the residual iron species, the precipitate was dissolved in dichloromethane (10 mL) and subjected to triple extraction with saturated citric acid solution (10 mL). In the end, the dry solid was extensively purified by column chromatography in the phase system—CH_2_Cl_2_:CH_3_OH, 50:1, *v*/*v*—to yield the targeted compounds **Pz1**–**Pz4**. Detailed spectral data of the synthesized porphyrazines can be found in the Appendix A.

### 3.2. Electrochemical Studies

Cyclic and differential pulse voltammetry studies were performed with Metrohm Autolab PGSTAT128N potentiostat/galvanostat, controlled by GPES (Eco Chemie, Utrecht, The Netherlands) and Metrohm Nova 2.1.4 software. The experiments were taken in a dichloromethane/0.1 M TBAP (both provided by Sigma-Aldrich, St. Louis, MI, USA) mixture previously degassed by nitrogen gas purging. A three-electrode system was employed—a glassy carbon (GC) working electrode (3 mm, area = 0.02 cm^2^), a Ag wire as a pseudo-reference electrode, and a platinum wire as a counter electrode. Before all measurements, the working electrode was polished with an aqueous 50 nm Al_2_O_3_ slurry (provided by Sigma-Aldrich) on a polishing cloth and was subsequently washed in an ultrasonic bath with a water–acetone mixture for 10 min in order to remove any kind of organic or inorganic impurity. The experiments were performed in room temperature using ferrocene (Sigma-Aldrich) as an internal standard. Therefore, the obtained results were adjusted to the ferrocenium/ferrocene couple (Fc^+^/Fc) potential.

### 3.3. Spectroelectrochemistry

The spectroelectrochemical studies were taken in a 1 mm path length quartz cuvette equipped with a Pt-gauze working electrode, a Ag/AgCl as a reference, and a platinum wire as an auxiliary electrode in a dichloromethane/0.1 M TBAP solution, previously deoxygenated by nitrogen gas purging. The proper applied overpotential was provided by Metrohm Autolab PGSTAT128N potentiostat/galvanostat controlled by Metrohm Nova 2.1.4 software. The UV–Vis absorbance spectra were recorded in the range of 300–1000 nm within 2 min (every 10 s), using an Ocean Optics USB 2000+ diode array spectrophotometer.

### 3.4. Oxidation Reaction Studies

The 1,3-diphenylisobenzofuran (DPBF) oxidation studies using synthesized iron(III) porphyrazines were performed in a 10 mm path length quartz cuvette placed in a holder at the top of a magnetic stirrer. The Ocean Optics USB 2000+ diode array spectrophotometer was employed to record the UV–Vis absorbance spectra of the reaction mixture during oxidation. The measurements were performed in dry DMF (TCI, Zwijndrecht, Belgium) stored over molecular sieves. The solvent was additionally deoxygenated prior to experiments by nitrogen gas purging. As well as 0.1 mM solutions of each iron(III) porphyrazine **Pz1**–**Pz4**, 0.185 mM solutions of 1,3-diphenylisobenzofurane were prepared. A total of 1.5 mL from each solution (both porphyrazine and substrate-containing) was placed in the quartz cuvette and stirred for 1 min. Next, the proper oxygen atom donor was added: (i) 20 μL of 11 mM *tert*-butyl hydrogen peroxide (TBHP) solution in DMF or (ii) 20 μL of 11 mM H_2_O_2_ solution in DMF. Thus, the oxygen atom donor/DPBF ratio of 2:1 was obtained in the reaction mixture in each sample. Next, the reaction mixtures were stirred for 10 min in the dark, eliminating the potential photodynamic effect that could occur upon light irradiation. The UV–Vis absorbance spectra were recorded for 8 min with 2 min intervals. In addition, control measurements without any oxygen atom donor (H_2_O_2_ or TBHP) or porphyrazine catalyst were performed for comparison.

## 4. Conclusions

Four iron(II/III) porphyrazines with pyrrolyl peripheral substituents were investigated in terms of their electrochemical activity, spectroelectrochemical electrolysis, and oxidizing activity in oxidation reactions. This study aimed to evaluate the impact of extending the macrocycle’s π electron system through the addition of phenyl rings to pyrrolyl moieties on the number of redox processes, the electrochemical HOMO–LUMO gap, the formation of anion and cation radicals, the efficiency of oxidation of 1,3-diphenylisobenzofuran as a reference substrate, and the susceptibility to radical-based degradation. Untypical behavior of the investigated porphyrazines was noticed in the cyclic and differential pulse voltammetry studies due to the presence of an extended π electron system in comparison to other porphyrinoid transition metal complexes. The occurrence of six well-defined redox couples assigned to Fe^2+^/Fe^3+^ transition and ring-based processes confirmed the thesis of increased electrochemical activity in porphyrinoids with bulky π-conjugated periphery. Analysis of the peaks indicated typical quasi-reversible one-electron transfer of all redox processes. The absence of weak electro-donating methyl groups replaced by aromatic moieties in **Pz3** and **Pz4** resulted in an increase in the calculated E_1/2_ gap between processes related to redox processes of the macrocycle ring by 0.1 V. For this reason, it can be presumed that further extension of the macrocycle’s periphery may lead to larger electrochemical band gaps, influencing the ability of the usage of such porphyrinoids as semiconductors. The significant results of the spectroelectrochemical studies were found in the case of **Pz2** and **Pz4** at the applied potential of −1.3 V. The appearance of new red-shifted bands during electrolysis at the negative potential can be explained by the formation of Fe^2+^ complex of porphyrazines. The investigated porphyrazines were resistant to the spectra changes at the positive potentials, which may indicate that the possible formation of cation radicals of ligands is hampered by the presence of additional phenyl substituents. In the oxidation activity assessment, performed with the use of reference compound 1,3-diphenylisobenzofuran and two oxygen atom donors in the dark, all investigated compounds revealed diverse efficiency. Due to the least bulky periphery, the most active porphyrazines were **Pz1** and **Pz2**. On the other hand, the increasing number of phenyl substituents in pyrrolyl groups in the case of **Pz3** and **Pz4** hampered the degradation of the catalyst during the reaction. Based on the obtained results, it can be assumed that the investigated porphyrazines may be considered for further experiments as systems capable of imitating the function of cytochrome P450 prosthetic groups in terms of oxidation and hydroxylation reactions, due to the structural similarity between the obtained porphyrazines and heme.

## Figures and Tables

**Figure 1 molecules-28-07214-f001:**
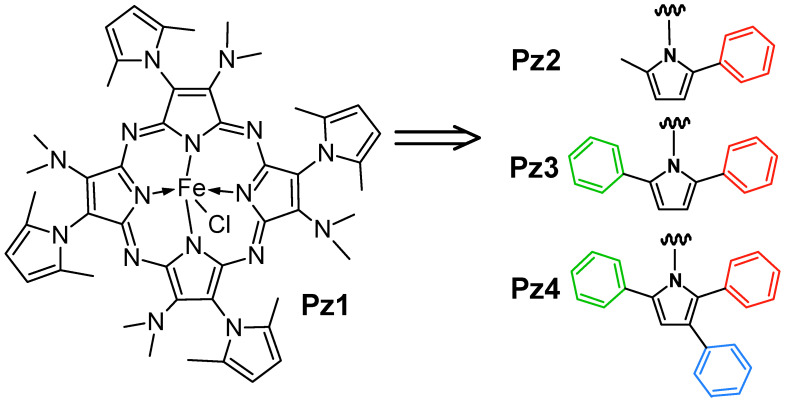
The chemical structures of investigated iron(III) porphyrazines **Pz1**–**Pz4**.

**Figure 2 molecules-28-07214-f002:**
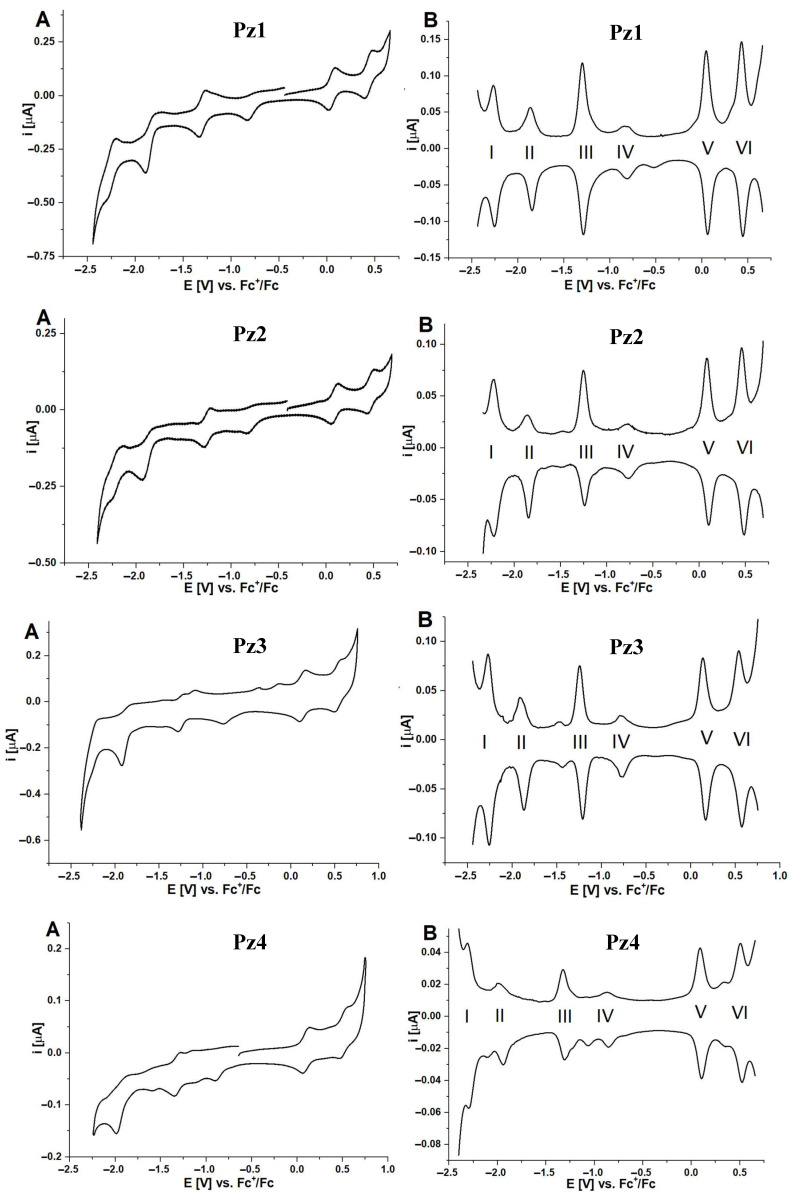
The cyclic (**A**) and differential pulse voltammograms (**B**) of investigated iron(III) porphyrazines **Pz1**–**Pz4** in dichloromethane/0.1 M TBAP. DPV parameters: modulation amplitude 20 mV; step rate 5 mV × s^−1^. CV scan rate 50 mV × s^−1^.

**Figure 3 molecules-28-07214-f003:**
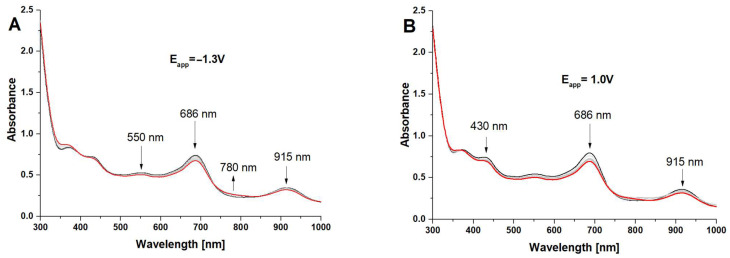
The UV–Vis spectra of **Pz2** in dichloromethane/0.1 M TBAP at E_app_ = −1.3 V (**A**) and at E_app_ = 1.0 V (**B**).

**Figure 4 molecules-28-07214-f004:**
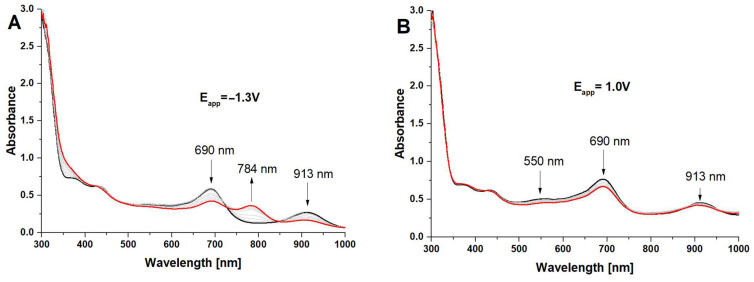
The UV–Vis spectra of **Pz4** in dichloromethane/0.1 M TBAP at E_app_ = −1.3 V (**A**) and at E_app_ = 1.0 V (**B**).

**Figure 5 molecules-28-07214-f005:**
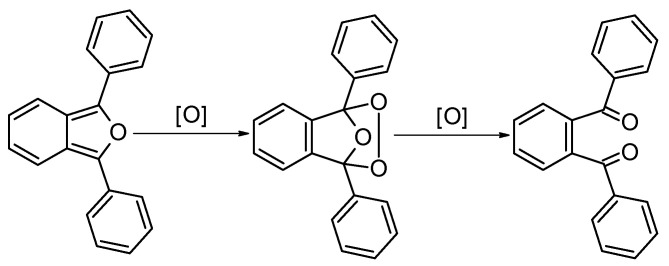
The schematic oxidation reaction of 1,3-diphenylisobenzofuran (DPBF).

**Figure 6 molecules-28-07214-f006:**
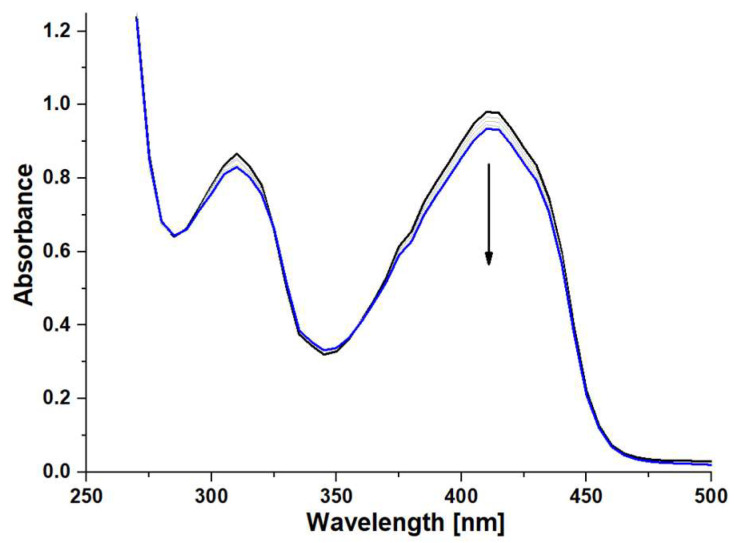
The UV–Vis spectra of DPBF in DMF recorded within 10 min of the oxidation process using **Pz1** as catalyst.

**Figure 7 molecules-28-07214-f007:**
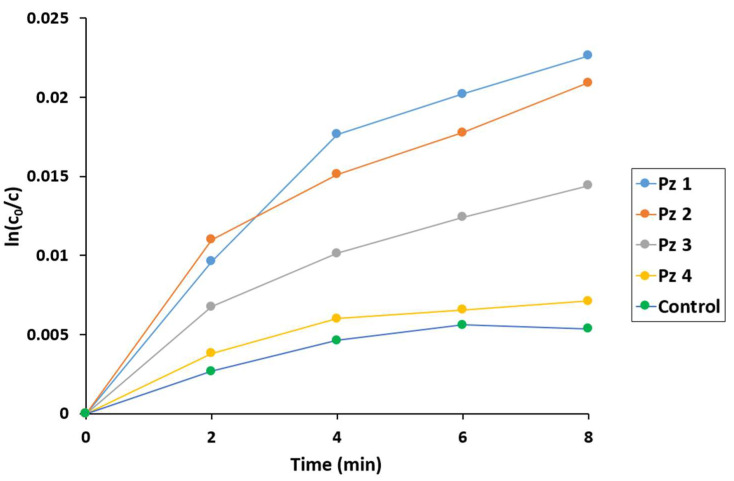
The logarithmic plot of DPBF concentration changes during the oxidation reaction using H_2_O_2_ as the oxygen atom donor.

**Figure 8 molecules-28-07214-f008:**
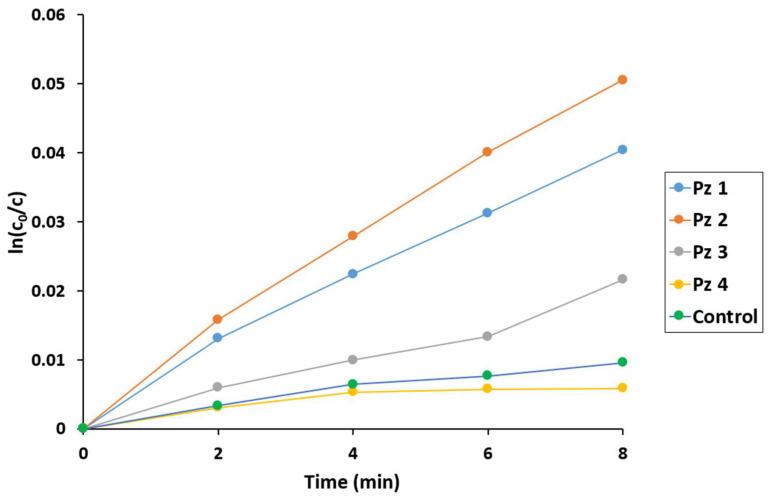
The logarithmic plot of DPBF concentration changes during the oxidation reaction using *t*-BuOOH as the oxygen atom donor.

**Table 1 molecules-28-07214-t001:** Electrochemical data of porphyrazines **Pz1**–**Pz4**.

		Fe(I)/Pz(−1)/ Fe(I)/Pz(−2) (I)	Fe(I)/Pz(0)/ Fe(I)/Pz(−1) (II)	Fe(I)/Pz(0)/ Fe(II)/Pz(0) (III)	Fe(II)/Pz(0)/ Fe(III)/Pz(0) (IV)	Fe(III)/Pz(0)/ Fe(III)/Pz(+1) (V)	Fe(III)/Pz(+1)/ Fe(III)/Pz(+2) (VI)	II—V E_1/2_ gap (V)
**Pz1**	E_1/2_ [V] vs. Fc^+^/Fc	−2.27	−1.87	−1.29	−0.84	0.06	0.44	1.93
	ΔEp [mV]	64	90	90	-	61	60	
**Pz2**	E_1/2_ [V] vs. Fc^+^/Fc	−2.23	−1.84	−1.23	−0.77	0.09	0.49	1.93
	ΔEp [mV]	60	140	87	-	62	50	
**Pz3**	E_1/2_ [V] vs. Fc^+^/Fc	−2.29	−1.89	−1.22	−0.77	0.15	0.55	2.04
	ΔEp [mV]	-	90	130	-	62	50	
**Pz4**	E_1/2_ [V] vs. Fc^+^/Fc	−2.33	−1.97	−1.31	−0.87	0.09	0.50	2.06
	ΔE_p_ [mV]	-	100	110	-	60	45	

CV scan rate 50 mV × s^−1^.

**Table 2 molecules-28-07214-t002:** Concentration changes in the substrate and catalysts during oxidation reactions using H_2_O_2_ and *t*-BuOOH as the oxygen atom donors.

Pz	H_2_O_2_ Δc/t × 10^7^(mmol/min)		*t*-BuOOH Δc/t × 10^7^(mmol/min)	
	DPBF	Pz Catalyst	DPBF	Pz Catalyst
**1**	6.7	17.0	12.0	8.6
**2**	6.2	18.0	15.0	8.5
**3**	4.3	7.3	6.4	4.9
**4**	2.1	2.6	1.8	2.0

## Data Availability

Not applicable.

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
