# Peer review of "The Influence of an Extended π Electron System on the Electrochemical Properties and Oxidizing Activity of a Series of Iron(III) Porphyrazines with Bulky Pyrrolyl Substituents"

_molecules, 2023, doi:10.3390/molecules28207214_

Round 1

Reviewer 1 Report

The submitted manuscript is devoted to physicochemical study of interesting series of iron(III) porphyrazines, in which structures one, two, or three phenyl groups were introduced onto auxiliary pyrrolyl substituents on the macrocycle.  The main aim is to understood deeper the effect of the increasing number of substituents on the electrochemical, spectroelectrochemical and catalytic properties of these interesting compounds.  It should be noted, synthesis of these compounds was already published by the authors in two previous articles, along with some physicochemical characterizations that included NMR, XRD, basic UV spectroscopic properties and Mössbauer Spectroscopy of iron complexes studying redox behavior and HOMO-LUMO gap. The results seem sound, and the content is of a good quality, which reflects expertise of the authors in this interesting area. However, the paper, as a whole, exhibits a bit incremental character and some parts suffer from several minor imperfections listed below.  

Figure 4: Authors would give precise values of UV peak maxima in this Figure or listed in an additional Table.

L 107: TBAP (tetrabutylammonium chlorate (VII)) would be better written as tetrabutylammonium chlorate

Catalytic reactions:  The use of term “catalysis” seems curious under conditions when ratio of catalyst to substrate was approximately 1: 2. Possibly, expressions such as activation would more appropriate. The new compounds would mimic action of heme in CYPs. I wonder, if heme could be involved in this study as reference, or an attention would be given to comparison with available literature data.

Conclusions seem wordy, and in several places only reproduce text from Results and discussion. 

Author Response

Dear Reviewer,

Please find the attached response to your comments.

Kind regards,

The Authors

Reviewer 2 Report

In this work, Koczorowski and Rebis presented a study of iron(II/III) porphyrazines with extending pyrrolyl peripheral substituents to understand the impact of introduced phenyl rings on the macrocycle’s electrochemical, spectroelectrochemical, and catalytic properties in oxidation reactions. The work is presented with enough experiments and a reasonable conclusion and is acceptable with minor revision.

1. The way the authors presented the structure of the molecules in Figure 1 is confusing. I recommend showing the structure of one of the molecules without R (instead of the left molecule with a smaller font and structure) and then the changing functional group.

2. The change in the signal of UV-Vis spectra is very minor in Figure 6. I recommend using a UV vis in the presence of Pz1 after a longer reaction time which makes the change more significant.

3. Did the authors record the NMR spectra of the reaction mixture to prove the oxidation of DPBF? Since one part of the study is the catalytic reaction, presenting more evidence, especially NMR spectra, to prove the reaction is required.

Author Response

(The authors gave the same response as above.)
